# Mechanotransduction events at the physiological site of touch detection

**Luke H Ziolkowski[1], Elena O Gracheva[1,2,3,4]\*, Sviatoslav N Bagriantsev[1]\***

[1]Department of Cellular and Molecular Physiology, Yale University School of Medicine, New Haven, United States; [2]Department of Neuroscience, Yale University School of Medicine, New Haven, United States; [3]Program in Cellular Neuroscience, Neurodegeneration and Repair, Yale University School of Medicine, New Haven, United States; [4]Kavli Institute for Neuroscience, Yale University School of Medicine, New Haven, United States

**Abstract** Afferents of peripheral mechanoreceptors innervate the skin of vertebrates, where they detect physical touch via mechanically gated ion channels (mechanotransducers). While the afferent terminal is generally understood to be the primary site of mechanotransduction, the functional properties of mechanically activated (MA) ionic current generated by mechanotransducers at this location remain obscure. Until now, direct evidence of MA current and mechanically induced action potentials in the mechanoreceptor terminal has not been obtained. Here, we report patch-clamp recordings from the afferent terminal innervating Grandry (Meissner) corpuscles in the bill skin of a tactile specialist duck. We show that mechanical stimulation evokes MA current in the afferent with fast kinetics of activation and inactivation during the dynamic phases of the mechanical stimulus. These responses trigger rapidly adapting firing in the afferent detected at the terminal and in the afferent fiber outside of the corpuscle. Our findings elucidate the initial electrogenic events of touch detection in the mechanoreceptor nerve terminal.

## Editor's evaluation

This fundamental work by Ziolkowski et al provides an exceptional advance in our understanding of the physiological sense of touch by directly performing in vivo patch-clamp recordings from vertebrate skin mechanoreceptor terminals. The provided evidence is compelling, overcoming a long-existing technical challenge and providing an experimental model to investigate the neuronal response to mechanical stimulation at the site of force detection that is of broad interest to physiologists, neuroscientists and biophysicists working on mechanoreceptors and mechanically activated ion channels.

**\*For correspondence:**
elena.gracheva@yale.edu (EOG);
slav.bagriantsev@yale.edu (SNB)

**Competing interest:** The authors declare that no competing interests exist.

## Introduction

In vertebrates, extrinsic touch is detected in the skin by cutaneous mechanoreceptors and somatosensory neurons of the peripheral nervous system. The afferent nerve fibers of these cells innervate the skin, where they form specialized ending structures which sense mechanical stimuli. Within the afferent terminals, mechanically gated ion channels (mechanotransducers), such as Piezo2, detect touch and transform it into mechanically activated (MA) current (*Handler and Ginty, 2021*). Extracellular recordings of mechanoreceptor afferents have previously revealed voltage changes originating from the terminals in response to mechanical stimulation, but the intracellular dynamics of these signals are not understood (*Loewenstein and Rathkamp, 1958*). As a result, direct evidence of mechanotransduction and MA current in the nerve endings of mechanoreceptors is lacking.

**Figure 1.** Mechanotransduction in the afferent terminal of the Grandry (Meissner) corpuscle. (**A**) Illustrated representation of the experimental setup. (**B**) The mechanical step stimulus applied with a glass probe (top), representative mechanically activated (MA) current responses in the terminal while voltage-clamped at –60 mV (middle), and simultaneous extracellular voltage signal from the connected afferent (bottom). (**C**) The mechanical stimulus (top), voltage responses and action potentials (APs) in the terminal in current-clamp (middle), and APs measured further along the afferent (bottom). (**D**) The current injection stimulus (top), voltage responses and action potentials in the terminal in current-clamp (middle), and APs measured in the afferent (bottom). (**E**) Example bright-field image of the experimental setup. (**F**) Quantification of the kinetics of MA current inactivation, (**G**) activation, (**H**) peak MA current-indentation relationship (n=7/6 afferent terminals for onset [ON]/offset [OFF], respectively), and (**I**) AP threshold measured in the dynamic ON phase of the stimulus and the dynamic OFF phase of the stimulus. Only the difference in inactivation $\tau$ between the ON and OFF phase was statistically significant (p<0.05). Statistics: Mann-Whitney U test (**F, G, and I**) or two-way ANOVA (**H**). Symbols indicate data from individual cells. Data in F–I were obtained from at least three independent skin preparations and shown as mean ± SEM.

The online version of this article includes the following source data for figure 1:

**Source data 1.** Original data for *Figure 1F–I*.

Studies of MA current and mechanotransducer biophysics have been limited to heterologous expression systems and dissociated somatosensory neurons *in vitro* (*Coste et al., 2010*; *Lewis et al., 2017*; *Schneider et al., 2017*; *Zheng et al., 2019*). Most notably, Piezo2, which mediates the detection of touch, displays fast-inactivating MA current in cultured cells and in dissociated neurons (*von Buchholtz et al., 2021*; *Chesler et al., 2016*; *Coste et al., 2010*; *Ranade et al., 2014*; *Wang et al., 2019*). However, it is unclear whether electrophysiological responses from somas of dissociated neurons accurately reflect that of MA current in the afferent terminal in the skin, due to potential differences in membrane geometry, level of ion channel expression, intracellular factors, and cellular/tissue environment between the two (*Richardson et al., 2022*). To our knowledge, intracellular recordings

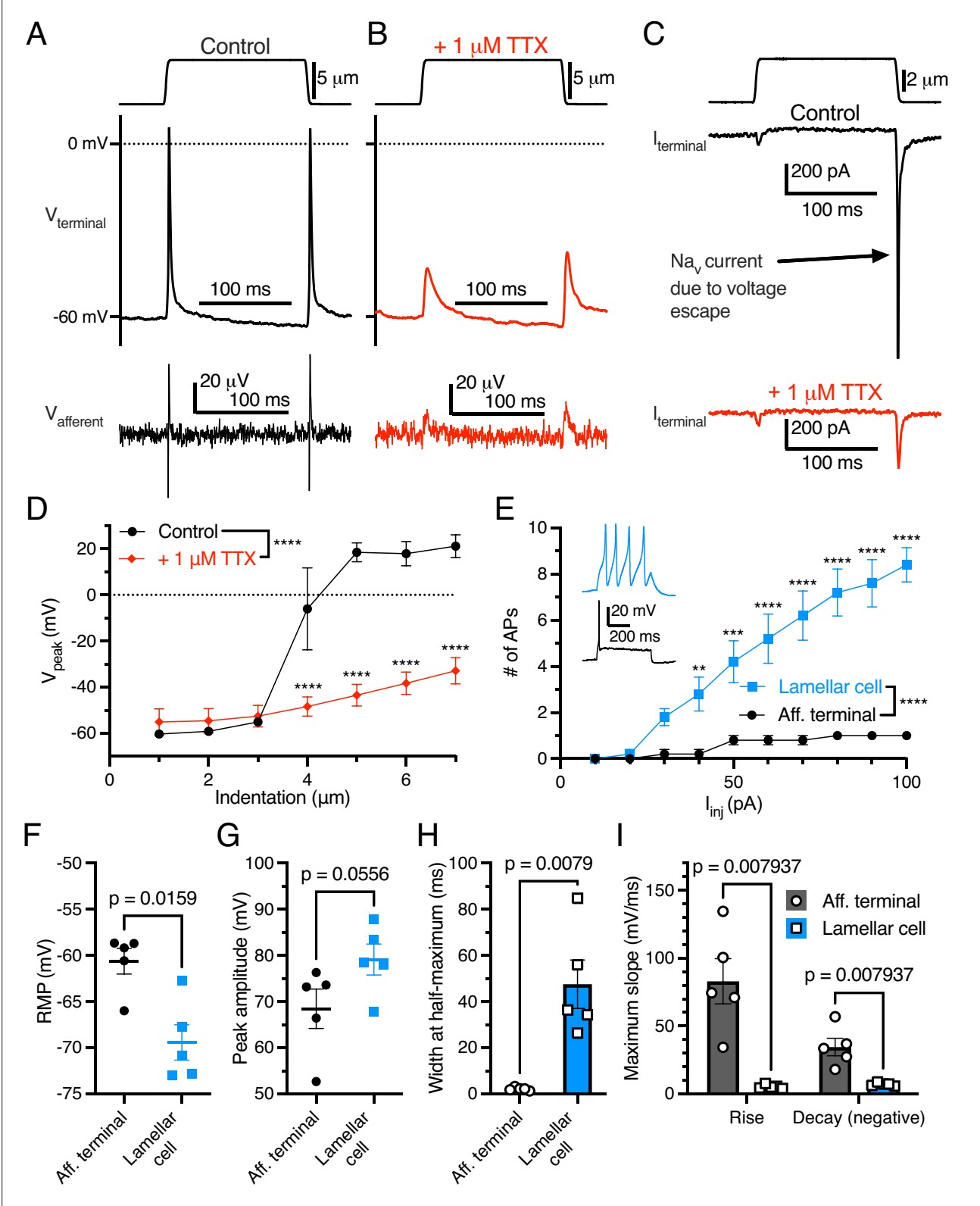

**Figure 2.** Electrogenic events in mechanoreceptor terminal and lamellar cells are carried out by different mechanisms. (**A**) A suprathreshold mechanical stimulus (top), action potentials (APs) in the terminal (middle), and propagated APs from the connected afferent (bottom). (**B**) A suprathreshold mechanical stimulus applied in 1 µM tetrodotoxin (TTX; top), AP-absent voltage responses in the terminal in current-clamp (middle), and extracellular receptor potentials in the afferent (bottom). (**C**) A suprathreshold mechanical stimulus (top), current responses in the terminal while voltage-clamped

*Figure 2 continued on next page*

*Figure 2 continued*

at –60 mV without 1 μM TTX (middle), and with 1 μM TTX (bottom). (**D**) Voltage-indentation relationship in the absence or presence of 1 μM TTX (n=5 for each group). (**E**) The number of APs from increasing current injections in lamellar cells and afferent terminals (n=5 for each group). Inset shows exemplar action potentials from a lamellar cell (blue) and afferent (black). (**F**) Resting membrane potential (RMP), (**G**) peak AP amplitude, (**H**) AP width at the half-maximum, and (**I**) the maximum slope of the AP rise or decay in the afferent terminal versus lamellar cells of the corpuscle. The AP-current injection relationship, RMP, width at half-maximum, max rise slope, and max decay slope were significantly different between the afferent terminal and lamellar cells (p<0.05). Statistics: Mann-Whitney U test (**F–I**) or two-way ANOVA with Holm-Sidak post-hoc test (**D and E**). **p=0.0084, ***p=0.0004, ****p<0.0001. Symbols indicate data from individual cells. Data in D–I were obtained from at least three independent skin preparations and shown as mean ± SEM.

The online version of this article includes the following source data for figure 2:

**Source data 1.** Original data for *Figure 2D–I*.

of mechanoreceptor terminals have not been previously reported due to the technical difficulties of accessing the axonal endings with patch-clamp electrodes. Consequently, the functional characteristics of mechanotransduction at the normal physiological site of touch detection remain unknown.

To address this gap in knowledge, we acquired patch-clamp recordings from the afferent terminals of Grandry (Meissner) corpuscles in the bill skin of the tactile specialist Mallard duck (*Anas platyrhynchos domesticus*). The Grandry corpuscle is an avian tactile end-organ innervated by rapidly adapting mechanoreceptors, which form thin terminals between Schwann cell-derived lamellar cells (*Nikolaev et al., 2020*; *Schneider et al., 2017*). The Grandry corpuscle's layered architecture, rapid adaptation, and sensitivity to transient touch make it structurally and functionally analogous to the mammalian Meissner corpuscle (*Neubarth et al., 2020*; *Schwaller et al., 2021*; *Ziolkowski et al., 2022*). Compared to mammals, the high density of corpuscles in the bill of tactile-foraging waterfowl enables persistent electrophysiological investigation of the afferent terminals in these end-organs, the results of which we report here.

## Results and discussion

We acquired patch-clamp recordings from the afferent terminal within the Grandry corpuscle using an *ex vivo* bill-skin preparation from late-stage duck embryos (*Figure 1A*). Mechanical stimulation of the voltage-clamped afferent terminal revealed fast-inactivating MA current only in response to the dynamic onset (ON) and offset (OFF) phases of the stimulus (*Figure 1B*). In current-clamp, both indentation with a probe (*Figure 1C*) and current injection (*Figure 1D*) caused depolarization of the membrane voltage, which initiated action potentials (APs) in the terminal during both phases. In three corpuscles in which the afferent terminal was patched, simultaneous single-fiber nerve recordings were also established using a section of the same afferent outside of the corpuscle (*Figure 1A and E*). In these cases, propagating APs from the afferent terminal were recorded in the afferent fiber with a one-to-one correlation to APs in the terminal (*Figure 1B–D*, bottom). When comparing the responses during the ON and OFF phases, we detected a difference between the rates of current inactivation (*Figure 1F*), but not the rates of activation, current-indentation relationship, or AP threshold (*Figure 1E–I*). The inactivation rate of MA current in the ON phase ($\tau$=8.95 ± 1.82 ms) in the terminal is notably similar to the inactivation rate of fast-inactivating MA current measured from the somas of murine and duck mechanoreceptors *in vitro* (*Coste et al., 2010*; *Schneider et al., 2017*; *Viatchenko-Karpinski and Gu, 2016*). Though duck Piezo2 also displays fast-inactivating MA current ($\tau$<10 ms at negative membrane potentials; *Schneider et al., 2017*), whether the ON phase MA current in the terminal is mediated by Piezo2 or another, unknown ion channel remains to be determined. Interestingly, the MA current seen during the OFF phase is a unique response not reported in dissociated neurons or expression systems, even though the OFF response is typical of rapidly adapting mechanoreceptors in *ex vivo* single-fiber recordings. The fast inactivation rate of the OFF response compared to the ON response implies a distinct or modified mechanism of mechanotransduction. This could potentially be dependent on the cellular structure or function of lamellar cells in the corpuscle.

As expected, the addition of tetrodotoxin (TTX) to the bill-skin preparation blocked APs and voltage-gated sodium current in the afferent terminal (*Figure 2A–D*). In some voltage-clamp experiments, mechanical stimulation resulted in large (>1000 pA) depolarizing currents (*Figure 2C*) which did not follow the expected current-indentation relationship (*Figure 1H*). These currents were blocked

by TTX and therefore were voltage-gated sodium currents resulting from a brief loss of voltage clamp, likely due to the complex geometry of the afferent.

Importantly, APs in the afferent terminal are physiologically distinct from APs fired by Grandry lamellar cells (*Figure 2E–I*). Lamellar cell APs are mediated by voltage-gated calcium channels, which are insensitive to TTX (*Nikolaev et al., 2020*). Lamellar cells fire multiple APs in response to large current injections, whereas the afferent terminal fires a maximum of one AP during the same stimuli (*Figure 2E*). Additionally, there were significant differences in resting membrane potential, AP width at half-maximum, and maximum slope of rise and decay between the two cell types. These results, along with the single-fiber afferent voltage data which mirrors the terminal voltage (*Figure 1B–D*), demonstrate that the recordings acquired here are unequivocally from the afferent terminal within the corpuscle.

Here, we have shown that mechanical stimulation evokes MA current in the afferent terminal which initiates propagating APs. Critically, MA current in the terminal has properties closely resembling those observed in dissociated neuron somas. This ultimately confirms the validity of using *in vitro* models to study mechanotransducers. At the same time, an important aspect of the afferent terminal response *in situ* is absent from cultured cells: the MA current in the OFF phase. Further studies of rapidly adapting corpuscles and other mechanoreceptor endings will be required to understand the mechanisms underlying both the OFF and ON responses. Together, these findings reveal fundamental characteristics of mechanotransduction at the physiological site of touch detection in mechanosensory neurons.

# Materials and methods

**Key resources table**

| Reagent type (species) or resource | Designation | Source or reference | Identifiers | Additional information |
|---|---|---|---|---|
| Biological sample | Duck bill skin (*Anas platyrhynchos domesticus*) | Metzer Farms | | Embryonic day E25-E27, Sex undetermined |
| Software and algorithm | pClamp 10 | Molecular Devices | RRID: SCR_011323 | |
| Software and algorithm | GraphPad Prism 9.4.1 | GraphPad Software, LLC | RRID: SCR_002798 | |

### *Ex vivo* bill-skin preparation

Experiments with duck embryos (*Anas platyrhynchos domesticus*) were approved by and performed in accordance with guidelines of the Institutional Animal Case and Use Committee of Yale University, protocol 11526. The bill-skin preparation was slightly modified from previously published methods (*Nikolaev et al., 2020*). Intact skin was carefully removed from the bill of duck embryos (aged embryonic day 25–27, sex not determined) using a sharp scalpel tip in ice-cold L-15 media. The bill-skin was placed upside-down (epidermis on bottom) in the recording chamber under a slice anchor. Corpuscles and afferents in the dermis were visualized on an Olympus BX51WI upright microscope with an ORCA-Flash 4.0 LT camera (Hamamatsu). At room temperature (22–23°C), the bill-skin preparation was treated for 5 min with 2 mg/mL collagenase P (Roche) in Krebs solution containing (in mM) 117 NaCl, 3.5 KCl, 2.5 CaCl$_2$, 1.2 MgCl$_2$, 1.2 NaH$_2$PO$_4$, 25 NaHCO$_3$, and 11 glucose, saturated with 95% O$_2$ and 5% CO$_2$ (pH = 7.3–7.4), then washed with fresh Krebs solution.

### Patch-clamp electrophysiology

Recordings were acquired at room temperature using a MultiClamp 700B amplifier, Digidata 1550 A digitizer, and pClamp 10 software (Molecular Devices). Standard-wall, 1.5 mm diameter borosilicate pipettes with tip resistances of 2–5 MΩ were pulled using a P-1000 micropipette puller (Sutter Instruments). Pipettes were filled with intracellular solution containing (in mM) 135 K-gluconate, 5 KCl, 0.5 CaCl$_2$, 2 MgCl$_2$, 5 EGTA, 5 HEPES, 5 Na$_2$ATP, and 0.5 Na$_2$GTP (pH 7.3 with KOH). All experiments were performed in Krebs solution at room temperature. Data were sampled at 20 kHz and low-pass

filtered at 2 kHz. Terminals were recorded in whole-cell mode and were held at –60 mV during voltage-clamp experiments. Resting membrane potential was measured in current-clamp mode shortly after breaking in. In both voltage- and current-clamp, mechanical stimuli were applied to a single corpuscle using a blunt glass probe (2–10 μm tip diameter) mounted on a piezoelectric-driven actuator (Physik Instrumente GmbH). A mechanical step stimulus was applied to corpuscles starting at 1 μm and increasing by 1 μm after each indentation. The static plateau of the step stimulus lasted 150 ms, while the ramp had a duration of 3 ms for both the ON and OFF phases. For both phases in each terminal, the inactivation rate ($\tau$) of the MA current was calculated by fitting a single exponential function ($I = I_0 \times \exp^{(-t/\tau)}$, where $I_0$ is the baseline-subtracted peak current amplitude, $t$ is the time from the peak current, and $\tau$ is the inactivation constant) to the decaying portion of the largest three MA current responses and averaging the fitted $\tau$ values. The activation $\tau$ was calculated similarly using the rise portion of the response (**Nikolaev et al., 2020**). The threshold was measured in current-clamp as the smallest indentation which elicited an AP. In current-clamp, depolarizing current steps (from 10 to 100 pA in 10 pA increments) were applied to elicit APs in the afferent terminal and lamellar cells. The first AP in these recordings was used to calculate the peak amplitude, width at half-maximum, and maximum slope of rise and decay for the terminal versus lamellar cells. Experiments were not corrected for liquid-junction potential.

## Single-fiber recording

Recordings from single afferent fibers of corpuscles were acquired simultaneously with patch-clamp recordings for three corpuscles, using the second channel of the MultiClamp 700B amplifier. Single-fiber recording pipettes were manufactured from thin-wall, 1.5 mm diameter borosilicate glass capillaries using a P-1000 micropipette puller (Sutter Instruments) to create tip diameters of 5–30 μm, then filled with Krebs solution. Pipettes were placed on an electrode headstage connected to a High-Speed Pressure Clamp (ALA Scientific Instruments). Light (1–20 mmHg) positive pressure was applied from the recording electrode to clear away tissue from a corpuscle-associated afferent. Negative pressure was then applied until a large section (~5 μm) of the afferent was sucked into the pipette. Extracellular afferent voltage was recording in current-clamp mode, sampled at 20 kHz and low-pass filtered at 1 kHz.

## Data analysis

Data from afferent terminals and lamellar cells were acquired from separate, individual preparations from different animals. Data were analyzed and plotted in GraphPad Prism 9.4.1 (GraphPad Software, LLC) as individual data points or means ± SEM unless otherwise indicated.

## Acknowledgements

We thank Dr. Yury Nikolaev for help with establishing the skin preparation, and members of the SNB and EOG laboratories for their contributions throughout the project. This study was partly funded by NSF grants 1923127, 2114084 (to SNB) and 1754286 (to EOG), and NIH grants R01NS097547 and R01NS126277 (to SNB).

## Additional information

### Funding

| Funder | Grant reference number | Author |
| --- | --- | --- |
| National Science Foundation | 1923127 | Sviatoslav N Bagriantsev |
| National Science Foundation | 2114084 | Sviatoslav N Bagriantsev |
| National Science Foundation | 1754286 | Elena O Gracheva |

| Funder | Grant reference number | Author |
|---|---|---|
| National Institutes of Health | R01NS097547 | Sviatoslav N Bagriantsev |
| National Institutes of Health | R01NS126277 | Sviatoslav N Bagriantsev |

The funders had no role in study design, data collection and interpretation, or the decision to submit the work for publication.

## Author contributions
Luke H Ziolkowski, Conceptualization, Data curation, Formal analysis, Validation, Investigation, Visualization, Methodology, Writing – original draft, Writing – review and editing; Elena O Gracheva, Conceptualization, Data curation, Supervision, Funding acquisition, Methodology, Writing – original draft, Project administration, Writing – review and editing; Sviatoslav N Bagriantsev, Conceptualization, Data curation, Supervision, Funding acquisition, Writing – original draft, Project administration, Writing – review and editing

## Author ORCIDs
Luke H Ziolkowski ⓘ http://orcid.org/0000-0002-3420-6782
Elena O Gracheva ⓘ http://orcid.org/0000-0002-0846-3427
Sviatoslav N Bagriantsev ⓘ http://orcid.org/0000-0002-6661-3403

## Ethics
Experiments with duck embryos (Anas platyrhynchos domesticus) were approved by and performed in accordance with guidelines of the Institutional Animal Care and Use Committee of Yale University, protocol 11526.

## Decision letter and Author response
Decision letter https://doi.org/10.7554/eLife.84179.sa1
Author response https://doi.org/10.7554/eLife.84179.sa2

# Additional files

### Supplementary files
• MDAR checklist

### Data availability
All data generated or analysed during this study are included in the manuscript and supporting file. Source data files have been provided for Figures 1 and 2.

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
