## [Editor Report]

This fundamental work by Ziolkowski et al provides an exceptional advance in our understanding of the physiological sense of touch by directly perfoming in vivo patch-clamp recordings from vertebrate skin mechanoreceptor terminals. The provided evidence is compelling, overcoming a long-existing technical challenge and providing an experimental model to investigate the neuronal response to mechanical stimulation at the site of force detection that is of broad interest to physiologists, neuroscientists and biophysicists working on mechanoreceptors and mechanically activated ion channels.

---

## [Decision Letter]

**Decision letter after peer review:**

Thank you for submitting your article "Mechanotransduction events at the physiological site of touch detection" for consideration by *eLife*. Your article has been reviewed by three peer reviewers, one of whom is a member of our Board of Reviewing Editors, and the evaluation has been overseen by a Senior Editor. The reviewers have opted to remain anonymous.

The individual reviews are provided below. Please note that the reviewers have pointed out some suggestions that may help to improve the manuscript. You may consider incorporating these suggestions to the final version for publication. Please take note of the points below and we hope you will continue to support eLife by reviewing for us and by submitting other papers going forward.

*Reviewer #1 (Recommendations for the authors):*

The manuscript by Ziolkowski et al. shows unprecedented in vivo recordings from skin mechanoreceptors. These recordings produce rapid excitatory currents (with fast activation and inactivation kinetics) when applying the mechanical stimulus (ON) and after removing it (OFF). This mechanically activated ionic current (MA) propagates through action potentials out of the terminals from a certain threshold. These action potentials are also sensitive to TTX, demonstrating a role of voltage-gated sodium channels in the process. In this study the authors overcome a long-existing technical challenge and present an experimental model to study skin mechanoreceptors. The contents seem adequate for a short report, and future work should warrant more detailed characterization of the ion channels involved in these MA currents (Piezo2 seems to be the clearest candidate due to all the literature on the subject).

This short report is very well written and structured. Suggestions that may help to improve the manuscript are:

– The activation and inactivation kinetics (Figure 1F-G) of the MA current, both ON and OFF, is fast and very similar, although for inactivation, the MA OFF current seems to be twice as fast as the ON current ( figure 1F). However, this difference is not clearly inferred from the recordings of panel 1B. The authors could consider including an inset directly comparing zoomed/scaled MA-ON and OFF currents, so this significant difference in the inactivation kinetics is made more evident.

– Figure 2E shows an example of the APs of the lamellar cell, blue line. Perhaps it would be interesting to accompany them with the AP that is generated in the afferent terminal (black line), which would be similar to what is shown in 1D. This would provide a comparison at-a-glance of some of the distinct properties of lamellar cell vs afferent terminals which are then summarized in 2G-I.

*Reviewer #2 (Recommendations for the authors):*

Overall, the experiments conducted by the authors were well designed and all claims and conclusions made were well supported by the data. Additionally, the experiments and data collected are required for a comprehensive understanding of touch detection which, up until now, has not been done. This is already a very strong manuscript with no major issues that need to be addressed. I only have one minor point to take into consideration.

The manuscript would improve if the significance of the study was better explained; i.e. why are these results so fundamental for our understanding of the sense of touch?

*Reviewer #3 (Recommendations for the authors):*

This short communication builds up on the idea that most electrophysiological recordings to study mechanoreceptors and the mechanically activated current responses are performed on the dissociated neurons soma and far away from the site of action. Here the authors performed elegant recordings from afferent terminal (within the Grandry corpuscles) as well as afferent fiber (single-fiber recordings), not done before to the best of our knowledge. This strategy is useful and establishes the fundamental characteristics of mechanotransduction at the physiological site of touch/mechanical detection in the mechanosensory neurons.

A few significant differences are found in the properties of mechanically activated responses recorded from lamellar cells and afferent terminal, suggesting distinct mechanism of action. Though the mechanisms are not explored in this study, this strategy provides the baseline for future electrophysiological studies for various mechanoreceptors.

Another interesting aspect of this study is the presence of electrical responses during the off phase (retrieval of mechanical stimuli). Usually in dissociated neurons and in other in vitro studies, mechanical stimuli evoke currents during the start of the stimulus. Here the authors were able to look into the unique aspect of the current responses during the ON phase and the OFF phase. The results are summarized elegantly and the data supports the claim, with substantial discussion points.

The work is complete in its current format, the data supports the claim. It's a nice sweet short paper, to the point. The authors didn't leave much room for suggestions but here are my two cents.

1) A little more analysis or a line or two on the kinetics of "off and on phase". This is an important finding and can be highlighted a bit more. By the look of the traces seems like the decay of current is fitted with a single exponential. Did you try fitting with two or more?

2) Action potential is recorded first time from these cell types. This point can be highlighted a bit more in the text.

3) Possibility of Piezo2 as the major MA channel or an unknown mechano channel can also be discussed.

---

## [Author Response]

Reviewer #1 (Recommendations for the authors):This short report is very well written and structured. Suggestions that may help to improve the manuscript are:– The activation and inactivation kinetics (Figure 1F-G) of the MA current, both ON and OFF, is fast and very similar, although for inactivation, the MA OFF current seems to be twice as fast as the ON current ( figure 1F). However, this difference is not clearly inferred from the recordings of panel 1B. The authors could consider including an inset directly comparing zoomed/scaled MA-ON and OFF currents, so this significant difference in the inactivation kinetics is made more evident.

We changed Figure 1B to a more representative trace where the slower inactivation in the ON phase compared to the OFF phase is more visually clear.

– Figure 2E shows an example of the APs of the lamellar cell, blue line. Perhaps it would be interesting to accompany them with the AP that is generated in the afferent terminal (black line), which would be similar to what is shown in 1D. This would provide a comparison at-a-glance of some of the distinct properties of lamellar cell vs afferent terminals which are then summarized in 2G-I.

In Figure 2E, we added an inset of afferent terminal AP response below lamellar cell AP response with added scale bar.

Reviewer #2 (Recommendations for the authors):The manuscript would improve if the significance of the study was better explained; i.e. why are these results so fundamental for our understanding of the sense of touch?

In abstract, we added "Until now, direct evidence of MA current and mechanically-induced action potentials in the mechanoreceptor terminal has not been obtained."

Reviewer #3 (Recommendations for the authors):The work is complete in its current format, the data supports the claim. It's a nice sweet short paper, to the point. The authors didn't leave much room for suggestions but here are my two cents.1) A little more analysis or a line or two on the kinetics of "off and on phase". This is an important finding and can be highlighted a bit more. By the look of the traces seems like the decay of current is fitted with a single exponential. Did you try fitting with two or more?

We used single-fit only because it was sufficient to capture the decay kinetics in our experiments.

2) Action potential is recorded first time from these cell types. This point can be highlighted a bit more in the text.

In abstract, we added "Until now, direct evidence of MA current and mechanically-induced action potentials in the mechanoreceptor terminal has not been obtained."

3) Possibility of Piezo2 as the major MA channel or an unknown mechano channel can also be discussed.

We added the following: “Though duck Piezo2 also displays fast-inactivating MA current (Schneider et al., 2017), whether the ON phase MA current in the terminal is mediated by Piezo2 or another, unknown ion channel remains to be determined."